# A Study of the Plan and Performance Evaluation Method of an 8-m^3^ Chamber Using Ventilation Experiments and Numerical Analyses

**DOI:** 10.3390/ijerph192013556

**Published:** 2022-10-19

**Authors:** Seonghyun Park, Seongwoo Park, Janghoo Seo

**Affiliations:** 1Department of Industry-Academic Cooperation Foundation, Kookmin University, 77, Jeongneung-ro, Seongbuk-gu, Seoul 02707, Korea; 2Department of Architecture, Graduated School, Kookmin University, 77, Jeongneung-ro, Seongbuk-gu, Seoul 02707, Korea; 3School of Architecture, Kookmin University, 77, Jeongneung-ro, Seongbuk-gu, Seoul 02707, Korea

**Keywords:** 8-m^3^ chamber, ventilation efficiency, computational fluid dynamics (CFD), carbon dioxide (CO_2_), performance evaluation

## Abstract

With increases in the time spent on indoor activities, the interests and technological demands regarding indoor air quality (IAQ) have also increased. Indoor air pollution is often caused by furniture or construction materials and chemical substances, such as volatile organic compounds (VOCs). As a way to remove such pollutants, efforts have been made to promote the management of indoor air quality through emission experiments. To conduct an experiment, such as the pollutant emission experiment involving substances harmful to the human body, a chamber to control various factors should be developed. By using such chambers, experimental variables can be minimized, quantitative analyses may be conducted, and the basic theory may be discussed. When the chamber is installed, it is not easy to change the existing installed conditions. Therefore, it is necessary to review feasibility with an accurate design. However, there is limited research on both how to quantitatively design the chamber and evaluate it. Therefore, this study investigates suitable chamber design methods and performance through ventilation performance evaluation to discuss potential development methods. In the chamber design step, a computational fluid dynamics (CFD) analysis was performed to estimate the ventilation efficiency according to the inlet and outlet positions to develop an 8-m^3^ chamber. Next, a ventilation experiment was performed using the tracer gas method for the performance evaluation, while the chamber interior airflow was simulated based on the CFD analysis. In a ventilation experiment using a tracer gas, the variation in gas density leads to concentration imbalance; as a result of concentration imbalance at each point, errors may occur in ventilation efficiency depending on the measurement point, causing the accuracy of the performance evaluation to fall. An attempt was made to resolve this problem by performing the ventilation experiment with a ceiling fan. The result indicated that the performance evaluation could be conducted without altering ventilation efficiency, coinciding with the CFD analysis result. Furthermore, when the concentration field was examined according to time in the CFD analysis, uniform concentration of chamber interior air allowed the ventilation efficiency to be calculated irrespective of the measurement point. Based on the findings, this study suggests a quantitative method of performance evaluation with an experiment in an 8-m^3^ chamber and a concurrent CFD analysis.

## 1. Introduction

With the increasing time spent on indoor activities, the interest and technological demands regarding indoor air quality (IAQ) have increased. Indoor air pollution is often caused by furniture or construction materials and chemical substances, such as volatile organic compounds (VOCs). As a way to remove such pollutants, efforts have been made to promote the management of indoor air quality through emission experiments, and test methods and standards for regulating the emission of pollutants from furniture or construction materials have been suggested. Caron et al. conducted experiments investigating VOCs emitted from wood particle board in a chamber. They found that VOCs diffusion related to VOCs mass transfer from a material’s surface to the surrounding air was the limiting step in VOCs emission for the solid material studied, and should therefore, be considered when developing ventilation strategies [1]. Zhixiang et al. researched catalysis for VOCs removal in technology and application [2]. Additionally, Shrubsole et al. reviewed IAQ guidelines for VOCs that were toxicological for the various pollutants [3]. The emission test could be conducted in FLEC (Field and Laboratory Emission Cell), which is a small device for measuring VOCs emissions. However, a large chamber could be used to conduct various experiments, such as particulate pollutants, including microorganisms, and removal efficiency using air purifiers in relation to IAQ. To conduct an experiment such as the pollutant emission experiment involving substances harmful to the human body, a chamber to control various factors should be developed. This is why it is important to construct an adequate chamber. When a chamber is installed, it is not easy to change the existing installed conditions. Therefore, it is necessary to review feasibility with an accurate design. However, there is limited research on how to both quantitatively design the chamber and evaluate it. This study aimed to investigate suitable chamber design methods, as well as performance, through ventilation performance evaluation and discuss potential methods. A ventilation experiment using a tracer gas was conducted for the ventilation performance evaluation, while a suitable tracer gas was selected by reviewing previous studies reporting tracer gas experiments [4,5,6,7].

In this study, CO_2_ was used as a tracer gas, which should be safe (non-toxic, non-allergenic, non-flammable), non-reactive (it should not react chemically or physically with the environment), and easily measurable (preferably it can be measured with low cost instrumentation which should be able to measure low concentrations). Moreover, the tracer gas should perfectly mix with air (similar density) and be distinguished from the constituents of air [8]. Alemida et al. performed a ventilation experiment using SF_6_ and CO_2_ tracer gases in a detached house condition to discuss suitable tracer gases. This previous study also showed that velocity was the most significant factor when the indoor or outdoor temperature was not clearly defined, with changes in ventilation rate occurring when the indoor to outdoor temperature gradient exceeded 3 °C [9]. CO_2_-based methods are convenient as CO_2_ is inert, its emission sources (people) are present in all buildings and usually well dispersed throughout occupied spaces, it is inexpensive, reasonably accurate in terms of measurement, and logging instruments are available. Occupant-generated CO_2_ has been widely used as a tracer gas for estimating ventilation rates [10]. Kabirikopaei et al. conducted a study on 220 classrooms in mid-western regions in the U.S. to estimate the rate of required ventilation based on data on CO_2_ generated by occupants of the classrooms. The team also claimed that the calculation accuracy for the required ventilation rate should be enhanced to better understand actual conditions, while indoor air quality should be evaluated [11]. On the utility of CO_2_, the American Society of Heating, Refrigerating and Air-Conditioning Engineers (ASHRAE) recommends that ventilation be evaluated and regulated, and indoor air quality be managed based on CO_2_ gas [12,13]. Batterman reviewed the potential use of CO_2_ as a tracer gas in the ventilation experiments of classrooms and stated that, based on the level of CO_2_ generated via the respiration of classroom occupants, the level of ventilation should be regulated according to the number and behavioral patterns of the occupants, while CO_2_ could prove to be useful in various studies regarding energy consumption in buildings [14].

Furthermore, a computational fluid dynamics (CFD) analysis for numerical interpretation can be performed and compared with the experimental results for a more accurate discussion of ventilation efficiency. In particular, a CFD analysis may be performed to measure indoor airflow and distribution of CO_2_ gas [15]. Using CO_2_ as the tracer gas, the concentration distribution is disproportionate in a given space, which may be quantitatively interpreted via a CFD analysis [16]. In addition, Khaoua et al. performed a numerical analysis according to wind velocity and direction to define the optimal ventilator locations in a greenhouse, using the data to locate and compartmentalize the vents [17]. As such, proposed designs can be reviewed in advance by contemplating ventilator locations with suitable levels of efficiency [18].

A chamber to achieve suitable target ventilation efficiency can be developed through a CFD-based numerical analysis, even in the chamber design step. The chamber should allow constant air density and pressure on the interior, in addition to the constant temperature and humidity conditions, while variables should be minimized towards a level of performance with controlled errors. For this, ventilation performance evaluation can be conducted to verify that the chamber has a suitable performance level. This study aimed to suggest a method for such performance evaluation. When the mixing is perfect, the course of concentration changes in the transient phase, and concentration levels at a steady state are completely described by a single system parameter, the nominal time constant [19]. Hence, for an accurate calculation of ventilation efficiency, the state of perfect mixing should be maintained throughout the experiment. To develop a chamber for such performance with controlled variables, accurate design and performance evaluation are required. However, there is a lack of reliable guidelines on specific design and performance evaluation methods to achieve a chamber with a controlled environment. In addition, few studies have been conducted regarding a ventilation experiment for performance evaluation using an 8-m³ chamber, which is relatively small in terms of required area and demands low installation cost, allowing small-scale home appliance experiments.

In this study, a CFD-based numerical analysis was performed in the designing step of an 8-m³ chamber to identify the inlet and outlet positions showing the most suitable ventilation efficiency. Next, considering the utility in construction, the chamber gas inlet and outlet were installed, and the ventilation experiment and CFD-based chamber performance evaluation were conducted. For the ventilation experiment, CO_2_ was used as the tracer gas. This could cause an imbalance in gas concentration due to the difference in density with air and possibly induce variations in concentration according to the measurement point and consequent errors in the calculation of ventilation efficiency. To resolve this concentration imbalance problem, a suitable performance evaluation method was suggested. The chamber design and evaluation methods were also discussed based on the CFD analysis simulating the 8-m³ chamber and the quantitative evidence. Based on the above descriptions, the roadmap of this study is shown in Figure 1 as follows.

## 2. Feasibility Study for Chamber Construction Plan

### 2.1. Construction of Chamber According to Ventilation Effectiveness Using Steady-State Analysis

#### 2.1.1. Definition of Ventilation Effectiveness

In this study, we assume indoor air is well mixed, as it is a common assumption for 8-m^3^ chambers. The methodology discussed in this study focuses on single zone systems: chambers. In addition, for the sake of simplicity, infiltration or exfiltration are not considered in this study on the assumption that the supple air is balanced with the return (exhaust) air; this is why chamber studies should be conducted in confidential conditions. In experimental ventilation studies using tracer gas CO_2_, equilibrium analyses are frequently used to calculate ventilation rates from indoor CO_2_ concentrations. This technique is based on a mass balance of CO_2_ in space. For a mechanically ventilated space, the mass balance of CO_2_ concentrations can be expressed as:(1)$$V\frac{dc}{dt}=Q\left(C_{0}-C\left(t\right)\right)+G\left(t\right)$$
where V is the space volume, C(t) is the indoor CO_2_ concentration at time t, Q is the volumetric airflow rate (fresh air) (into and out of), $C_{0}$ is the outdoor CO_2_ concentration, and G(t) is the CO_2_ generation rate at time t. If we assume Q, $C_{0}$, and G(t) are constant, Equation (1) can be solved as follows:(2)$$C\left(t\right)=C_{0}+\frac{G\left(t\right)}{Q}+\left(C\left(t\right)-C_{0}-\frac{G\left(t\right)}{Q}\right)e^{-It}$$
where C(0) is the indoor CO_2_ concentration at time 0, I = *Q*/V, air change rate. It has a unit of time, which is a reciprocal of the nominal time constant. The air change rate is a dimensionless number divided by ventilation time [20]. Equation (2) is used as an essential tool to dynamically calculate the ventilation rate in this study when C(t), $C_{0}$, G(t), and t are known. If the CO_2_ generation rate is constant for a sufficient time, the last term on the right side of Equation (2) converges to zero, and the equation can be rewritten as:(3)$$C_{eq}=C_{0}+\frac{G\left(t\right)}{Q}$$
where $C_{eq}$ is called the equilibrium CO_2_ concentration. Equation (3) is referred to as the equilibrium analysis, as previously mentioned [21]. This equation can be used as an indicator when performing a ventilation experimental study.

#### 2.1.2. Ventilation Effectiveness According to the Age of Air

In designing an 8-m³ chamber in this study, a steady-state CFD analysis was performed to construct a chamber exhibiting the most outstanding ventilation efficiency. In addition, chamber performance was evaluated through a ventilation experiment based on the concentration decay method using CO_2_ as the tracer gas; the result was compared with the CFD analysis result. For comparison, ventilation efficiency may be evaluated using the concept of the age of air [22]. In the case of limited devices and experimental setup, an unsteady-state CFD analysis may be performed to accurately simulate the tracer gas experiment and age of air [23]. Thus, whether a numerical analysis could simulate a ventilation experiment in an 8-m^3^ chamber to a certain level or above was verified. For this, the room mean age of air (RMA) was calculated and then used to estimate the ventilation efficiency.

The room mean age of air can be obtained as the area of the time-dependent concentration decay curve as the integration of the concentration decay according to time [24]. Therefore, it can be formulated in Equation (4), as follows. Hence, a ventilation experiment using an 8-m³ chamber was performed, and the concentration decay was examined. Next, the concentration decay curve was used to calculate the room mean age of air, and based on the consequent ventilation efficiency, the chamber performance was evaluated.
(4)$$\tau_{p}=\frac{\int_{0}^{\infty }\;t\times C_{e}\left(t\right)dt}{\int_{0}^{\infty }C_{e}\left(t\right)dt}$$

#### 2.1.3. Ventilation Effectiveness of the 8-m^3^ Chamber According to Position of Inlet and Outlet Using Steady-State Analysis

Prior to the installation of a large-scale 8-m³ chamber, a commercial CFD program, Fluent, was used to perform a numerical analysis to review the inlet and outlet positions. Five cases were set with a fixed inlet position and varying outlet positions, as illustrated in Figure 2. Then, the boundary conditions for all five cases were set, as presented in Table 1, and the CFD analysis was performed in the condition of a 1.0 air change per hour (ACH) [1/h]. The turbulence model was validated as described in prior research and as follows; Fang et al. compared the measured and the simulated air temperatures and velocities at the four different locations using three different available turbulence models: SST k-ω model, Standard k-ε model, and RNG k-ε model, respectively. Results of the SST k-ω model performed better than the other two in terms of simulating both air temperature and velocity profile in chamber [25]. Kobayashi et al. also investigated four turbulence models. By comparing all the results obtained from the experiment and numerical calculations, it was concluded that the turbulence model of SST k-ω model has a sufficient accuracy to analyze the target room [26]. As such, the SST k-ω turbulence model was adopted because it showed high reliability in the room.

Table 2 presents the analysis results of the changes in chamber interior airflow using the steady-state CFD analysis, calculating the room mean age of air and ventilation efficiency. Comparing the five cases according to the inlet and outlet positions showed that ventilation efficiency for a suitable level of performance was the highest for Case 1, followed by Cases 5 and 2. This may indicate that the target ventilation efficiency has been achieved, as the fresh air injected through the inlet was adequately mixed with the chamber’s interior air before leaving the chamber. For Cases 3 and 4, the ventilation efficiency decreased, presumably because the fresh air through the inlet did not adequately mix with the chamber interior air before leaving through the outlet. Hence, the large-scale 8-m³ chamber was designed based on Case 1 and the 8-m³ chamber is shown as Figure 3.

## 3. Comparison of Experiments and Numerical Analysis for 8-m^3^ Chamber Evaluation

### 3.1. CFD Boundary Conditions for Ventilation Experimental Study

#### 3.1.1. Species Transport Equation for Performance Evaluation of Chamber

To simulate a ventilation experiment using CO_2_ as the tracer gas, the species transport equation was applied. The chemical reactions are incorporated inside the chamber by selecting the species transport model. Various species, e.g., CO, H_2_, CH_4_, CO_2_, N_2_, O_2_, H_2_O, and C(s), were included in the model. The chemical reaction follows the conservation equation for each species’ convection, diffusion, and reaction.

The standard form of species transport equation is defined as:(5)$$\nabla \cdot \;(\rho \vec{v}Y_{i})=-\nabla \;\cdot \;\vec{J_{i}}+R_{i}+S_{i}$$
where diffusion flux $\vec{J_{i}}$ for turbulent flow is given by:(6)$$\vec{J_{i}}=-\left(\rho D_{i,m}+\frac{\mu_{t}}{S_{c_{t}}}\right)\nabla Y_{i}-D_{T,i}\frac{\nabla T}{T}$$

Species transport equations are solved individually for each species in Ansys fluent. In this equation, $Y_{i}$ is the mass fraction of species. The subscript i represents the species (i.e., CO, H_2_, CH_4_, CO_2_, N_2_, O_2_, and H_2_O), and $R_{i}$ denotes the net rate of production of species i by chemical reaction. $D_{i,m}$ is the mass diffusion coefficient, and $D_{T}$ is the turbulent diffusivity. The source term $S_{i}$ describes the destruction or creation of species due to chemical reactions. Turbulent Schmidt number $S_{c_{t}}$ is 0.7, which is the turbulent viscosity ratio to eddy diffusivity ratio [27].

#### 3.1.2. Velocity Profile of Inlet

In a chamber environment where both temperature and humidity are controlled, the airflow introduced through the inlet is the most significant factor in a ventilation experiment. Thus, the velocity profile was produced to set the pattern of flow and velocity at the inlet as the boundary conditions for CFD analysis. To produce the velocity profile, 8 m³/h of flow rate was applied corresponding ACH of 1.0 [1/h] for the 8-m³ chamber, and measurements were taken. As Figure 4 shows, an air velocity meter was installed at 21 measurement points, at which data were collected in 5-min intervals. 

For CFD analysis, the inlet pipe supplying the air to the chamber was modeled, and the pipe flow was analyzed. Next, the model showing the fewest errors across the 21 points and the most similar pattern to the measured values at each point was reviewed per turbulence model. The turbulence model was selected with the highest accuracy model compared to actual measurements, and the steady-state method was used for analysis by creating approximately 88,000 grids.

Based on measurements for the velocity profile formed according to the injected flow ACH of 1.0 [1/h], the boundary conditions for the velocity profile with the highest similarity to the numerical values in the CFD analysis were examined. As a result, the conditions for the velocity profile with a similar pattern to that shown in Figure 5 were determined to be applicable. In line with this, the velocity at each of the 21 measurement points was compared. Figure 6 shows the comparison of the standard k-ε turbulence model and the SST k-ω turbulence model used to determine which one had better accuracy. The standard k-ε turbulence model was chosen because it showed greater accuracy on each velocity of the points. The relative error rate was within 5% at all points, except for the four close to the walls. Thus, the velocity profile with the highest trend was used in the CFD analysis to increase the accuracy of the numerical analysis.

#### 3.1.3. Boundary Condition Using a Fan

For a case study using the ceiling fan, the measured values were compared with the CFD analysis data. The airflow rate at a specific point was measured at 900 revolutions per minute (RPM) and at 1800 RPM for the operation of two ceiling fans in an actual chamber. In line with this, the airflow rate at the identical point was compared with the model of the CFD fan condition. The measurement points were at 300 mm and 1500 mm from the floor based on the center of each fan, and the velocity was measured for approximately 10 min. The locations are as shown in Figure 7.

Using the mean velocity at the measurement points, the pressure jump regarding the CFD fan boundary condition was estimated. Compared to the measured values, the CFD calculation accuracy at 900 RPM yielded a 4% relative error at Point 1, and a 2% relative error at Point 3, as shown in Figure 8. The airflow was low at Points 2 and 4, as the mean velocity was ≤0.15 m/s; the error rate was interpreted as low. In addition, the accuracy at 1800 RPM was 2.5% relative error at Point 1, and 1.8% relative error at Point 3, compared to the measured values, as shown in Figure 9. The accuracy of the numerical analysis could be increased through comparison against the changes in airflow with the operation of a fan, by which the airflow conditions on the chamber interior were simulated.

### 3.2. Experimental Study in the 8-m³ Chamber

To suggest a guideline and quantitative evidence regarding the measurement points in the ventilation experiment for performance evaluation, the ventilation experiment was performed with the CFD-based numerical analysis. The instrument was installed at 400 mm, 1000 mm, and 1700 mm from the floor of the 8-m³ chamber, as shown in Figure 10. Throughout the ventilation experiment, the outdoor CO_2_ concentration was 490 ppm, while the chamber temperature was maintained at 25 °C. The experimental conditions were set to examine concentration decay from the initial concentration of 7497 ppm of the injected CO_2_ tracer gas.

After the injection, the fan was used to adequately mix the chamber interior air and tracer gas for 20 min to form a state of complete mixing. Figure 11 shows the change in concentration from CO_2_ injection to fan operation at each point. The complete mixing state was reached within about 20 min after operating the fan. The operation of the fan was halted, and fresh air was injected 1.0 ACH to perform the ventilation experiment. In addition, concentration fields and airflow that cannot be examined in an actual experiment were studied in the CFD analysis with identical boundary conditions.

Figure 12 shows the results of the ventilation experiment in 1.0 ACH conditions, indicating the lack of significant variation compared to the rate of concentration decay, calculated by Equation (2). As a result, the level of performance of the 8-m³ chamber used in this study was determined to be suitable for ventilation experimentation. Subsequently, the accuracy of CFD analysis according to the operation of the fan was compared, and a case study was conducted in accordance with the fan RPM. 

## 4. Results and Discussions

### 4.1. Comparison Measurements and CFD

With identical boundary conditions to the previous ventilation experiment, a CFD analysis was performed. At identical measurement points, data were collected in 10-sec intervals to draw the concentration decay graph, and the mean age of air was compared at localized points. As can be seen in Figure 13, the relative error between the experiment and the CFD analysis at each measurement point was 9%, 1%, and 4% at the point closest to the floor toward the point farthest from the floor, based on the mean age of air at localized points. The error rate indicated that, despite the points being localized, the accuracy was high with the variation in room mean age of air <1%. The CFD analysis was shown to simulate the ventilation experiment with high accuracy when the velocity profile was produced with high accuracy in the controlled chamber environment, while the reliability of the results was shown to be high.

The result of a CFD analysis that precisely simulates the chamber environment by time to examine the chamber interior concentration field is shown in Figure 14. An imbalance in the concentration field was detected as time passed, caused by the gradual downward movement of the high-density gas over time, due to the density difference between CO_2_ and air.

In this study, the imbalance in the concentration field due to the density difference between air and CO_2_ as the tracer gas in the ventilation experiment was examined in the CFD analysis. What the results indicated was the potential variation in the room mean age of air according to the location of the instrument. Furthermore, an error in the calculation of room mean age of air could prevent the calculation of accurate ventilation efficiency. Even if several instruments were installed, the locations could not be specified; the result of which would be an inaccurate performance evaluation.

### 4.2. Comparison Measurements and CFD Depending on the Operation of the Fan

As previously described, the measurement points could not be specified based on chamber conditions in a ventilation experiment using a tracer gas with a density difference from air. Therefore, this study performed the ventilation experiment with the operation of the fan with the case study based on the fan RPM. Three cases were set for the ventilation experiment: the case with the fan at 900 RPM; the case with the fan at 1800 RPM; and the case without the operation of the fan. The results were compared with numerical analysis. The measurement time throughout the experiment was set to 30 min, as it allowed the trend of significant concentration decay to be examined at half of the nominal time constant. As shown in Figure 15, the results indicated that ventilation efficiency was almost identical in the three cases. Thus, it was shown that the operation of the fan did not affect ventilation efficiency, and fan RPM could not be a variable in a ventilation experiment.

A case study was subsequently performed in a CFD analysis with identical conditions to the experiment. As shown in Figure 16, the rate of concentration decay was almost identical between the experiment and the CFD analysis. The analysis with the operation of the fan also produced almost identical results. Table 3 presents the ventilation efficiency in each case and the relative error compared to the experiment. In all cases, the relative error was within 4%, which verified that the CFD analysis could simulate the experiment with high accuracy. In addition, as shown by the experiment, ventilation efficiency was not affected by the operation of the fan or the fan RPM. This implied that, for performance evaluation, ventilation experiments could be performed with the operation of the fan without any problems.

In this study, the boundary conditions for CFD analysis that can simulate the ventilation experiment with high accuracy were predominantly investigated. Then, the imbalance of the time-dependent concentration field was examined in an unsteady-state numerical analysis. Next, after verifying the lack of variations in the ventilation efficiency, based on the presence or absence of the fan, the effect of the fan operation was examined through CFD analysis. The results showed a uniform distribution of concentration fields in the ventilation experiment with the operation of the fan, as shown in Figure 17. This implied that the problem of concentration imbalance caused by the density difference could be resolved by performing the ventilation experiment with the operation of the fan, regardless of the fan RPM, with a negligible impact on ventilation efficiency. The results suggested that the operation of the fan had no effect on the results of the ventilation experiment in performance evaluation.

In addition, a significant concentration decay was found when concentration variation was examined at the outlet positions most frequently used for measurements to calculate the spatial mean age of air in a previous study. Figure 18 shows the room mean concentration decay and the outlet positions according to the fan operation conditions. The room mean age of air could be measured at a single position regardless of the measurement points if the ventilation experiment could be performed with the operation of the fan. The room mean age of air could also be calculated without the calculation of the first moment of area at the outlet position.

## 5. Conclusions

This study investigated the ventilation efficiency according to the inlet and outlet positions in a large-scale 8-m³ chamber, applying the results in the chamber design. A performance evaluation was conducted in the subsequent ventilation experiment using a tracer gas in the 8-m³ chamber. The most widely used tracer gas is CO_2_; however, an imbalance in the concentration field due to the density difference with air unavoidably occurs with CO_2_. This caused errors in ventilation efficiency according to measurement points. A way to resolve such errors would be to use data from three or more instruments to obtain the mean, but there was insufficient evidence to determine the measurement points. To resolve these problems, the ventilation experiment in this study was performed with the operation of the fan; the results are as follows:Ventilation experiments could be performed without significant variation in ventilation efficiency, even with the operation of a fan.Ventilation experiments using a tracer gas with the operation of a fan solved the problem of concentration imbalance caused by the density difference between the gas and air, as a state of complete mixing was maintained throughout the experiment.As the concentration field was maintained uniform, the room mean age of air (RMA) could be estimated irrespective of location, even with the installation of a single instrument.

A problem in the conventional ventilation experiment using a tracer gas was pointed out, and a method to resolve the problem was suggested. In addition, by performing the ventilation experiment, the chamber performance was evaluated, and a method of performance evaluation was suggested. Thus, by combining the ventilation experiment and CFD analysis in an 8-m³ chamber, quantitative evidence for the performance evaluation was presented, and its potential use was verified. The 8-m³ chamber used in this study satisfied the target ventilation performance, thereby being verified as a chamber that can ensure the performance in homes or small-scale standard tests. In addition, among large-scale chambers with mechanical ventilation and suitable air tightness, the 8-m³ chamber is advantageous with a small area and relatively low installation cost. A follow-up study will be conducted with a glove box allowing an experiment with a UV lamp for sterilization and microorganisms, for which the design and development are complete. Through the use of the box, fine particulate matter and microorganisms could be generated to conduct an experiment to control indoor air quality.

## Figures and Tables

**Figure 1 ijerph-19-13556-f001:**
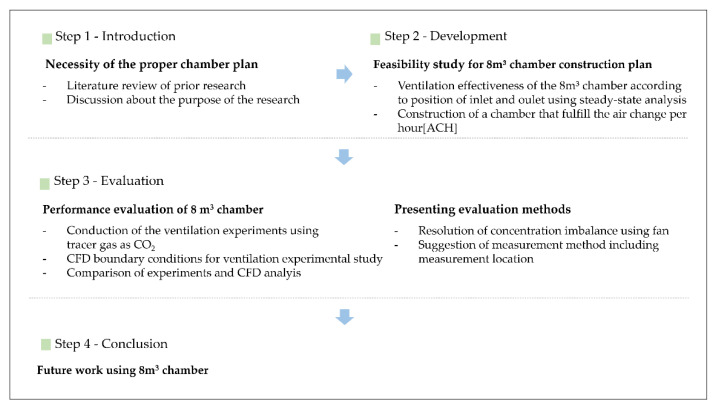
Roadmap of this research.

**Figure 2 ijerph-19-13556-f002:**
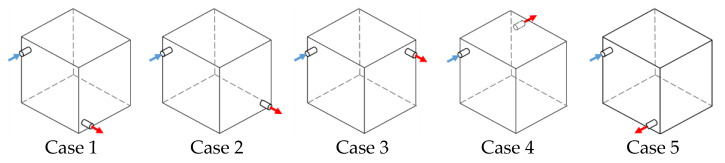
Case of the ventilation effectiveness according to position of inlet (blue) and outlet (red).

**Figure 3 ijerph-19-13556-f003:**
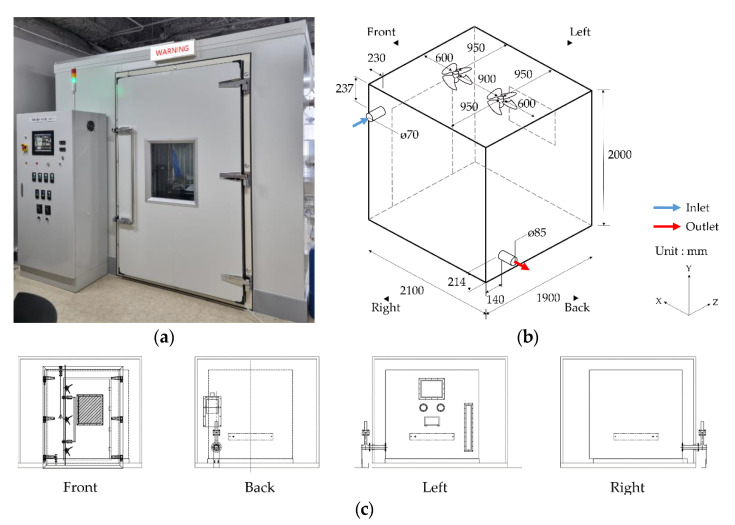
8-m^3^ chamber considering the ventilation effectiveness: (**a**) installed chamber; (**b**) schematic of 8-m^3^ chamber; (**c**) elevations of chamber.

**Figure 4 ijerph-19-13556-f004:**
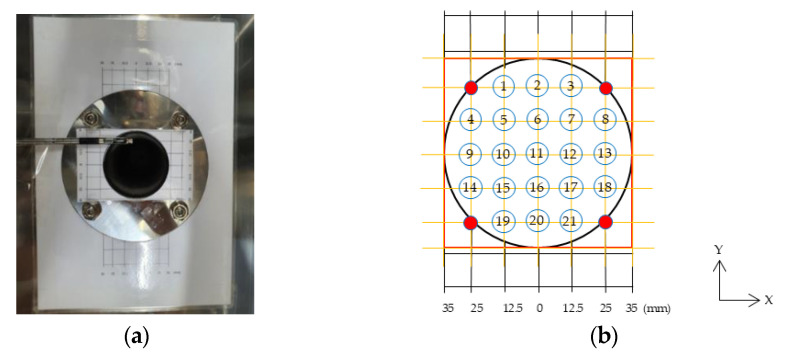
Measurements of the 21 locations velocity: (**a**) inlet of chamber; (**b**) the points of 21 measurements locations.

**Figure 5 ijerph-19-13556-f005:**
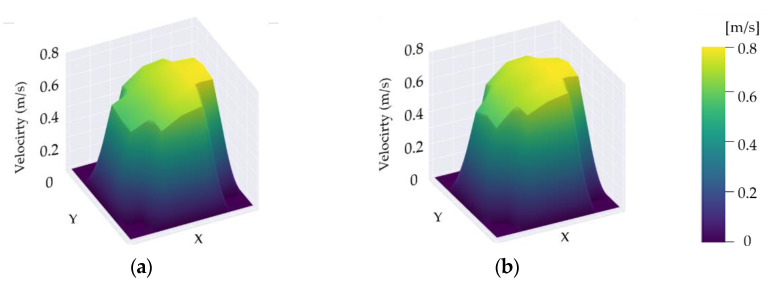
Comparison of velocity profile: (**a**) measurements value; (**b**) CFD value.

**Figure 6 ijerph-19-13556-f006:**
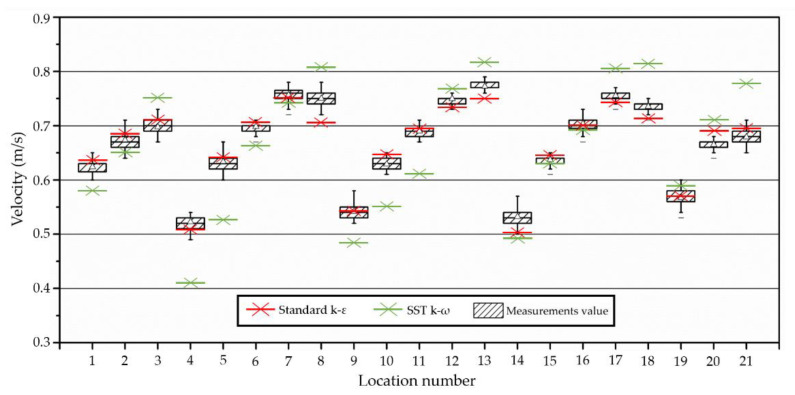
Relative error between a CFD value and measurements value of each 21 points; comparison of standard k-ε turbulence model and SST k-ω turbulence model.

**Figure 7 ijerph-19-13556-f007:**
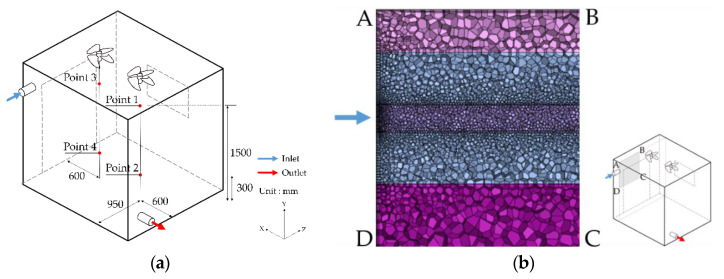
Measurements points: (**a**) schematic of chamber with fan; (**b**) mesh condition of section ABCD.

**Figure 8 ijerph-19-13556-f008:**
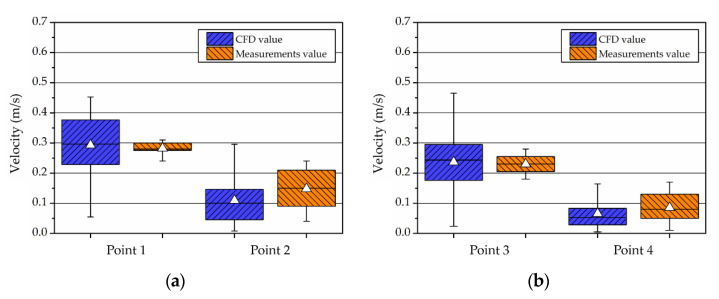
Relative error of CFD about measurements value in 900 RPM case: (**a**) fan 1; (**b**) fan 2.

**Figure 9 ijerph-19-13556-f009:**
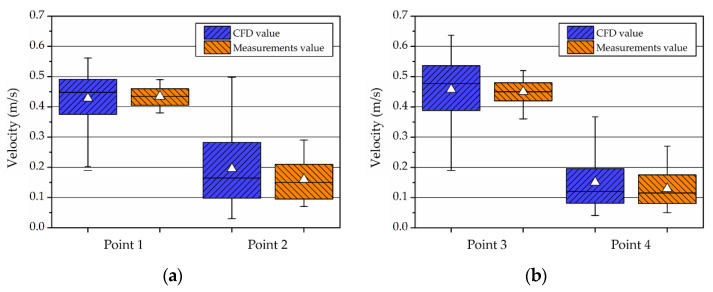
Relative error of CFD about measurements value in 1800 RPM case: (**a**) fan 1; (**b**) fan 2.

**Figure 10 ijerph-19-13556-f010:**
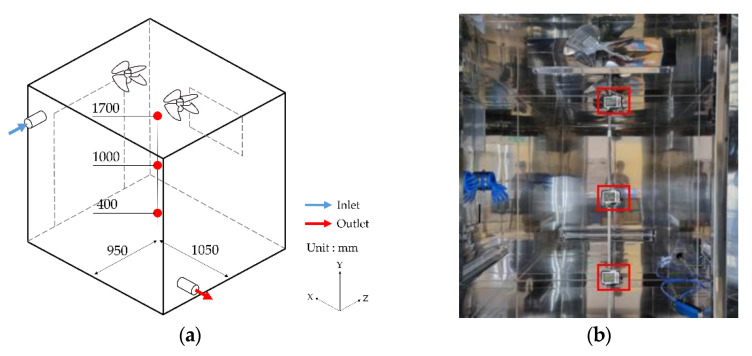
Ventilation experiments using tracer gas as CO_2_: (**a**) measurement points of experimental study; (**b**) measuring instrument location and experimental condition of 8-m³ chamber.

**Figure 11 ijerph-19-13556-f011:**
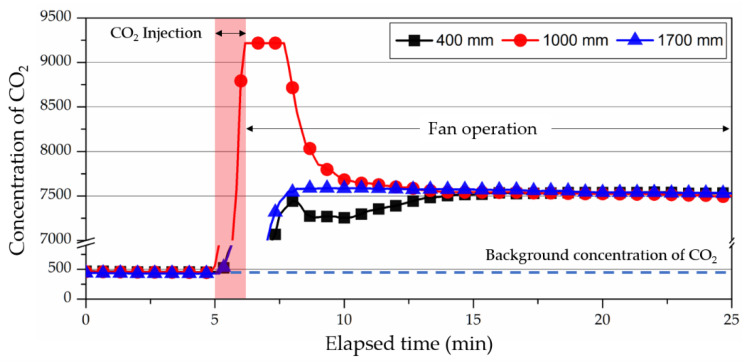
Elapsed time of reaching complete mixing after fan operation.

**Figure 12 ijerph-19-13556-f012:**
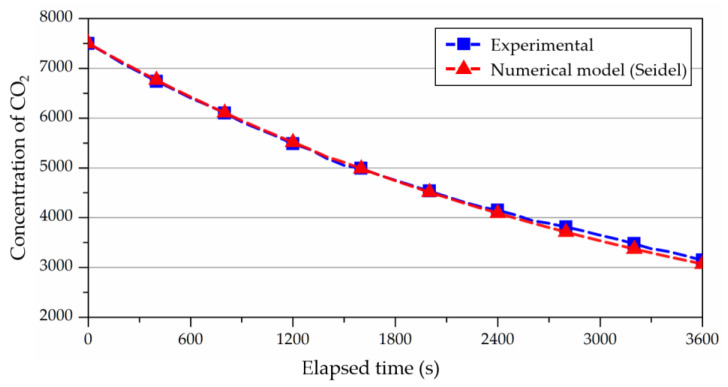
The results of the ventilation experiments using tracer gas as CO_2_.

**Figure 13 ijerph-19-13556-f013:**
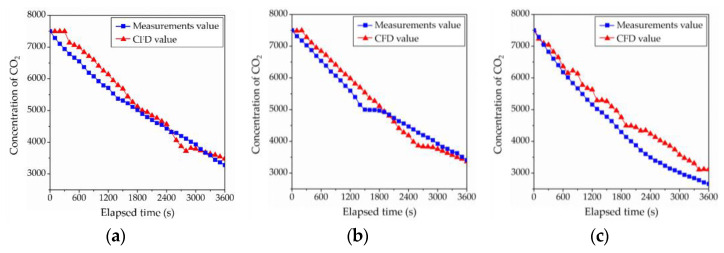
Comparison of the CFD and measurements value: (**a**) concentration decay of 400 mm point; (**b**) concentration decay of 1000 mm point; and (**c**) concentration decay of 1700 mm point.

**Figure 14 ijerph-19-13556-f014:**
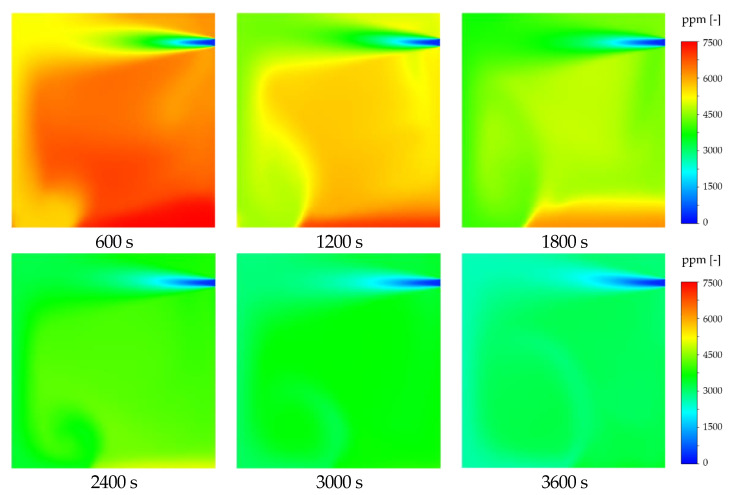
Change of concentration field according to elapsed time.

**Figure 15 ijerph-19-13556-f015:**
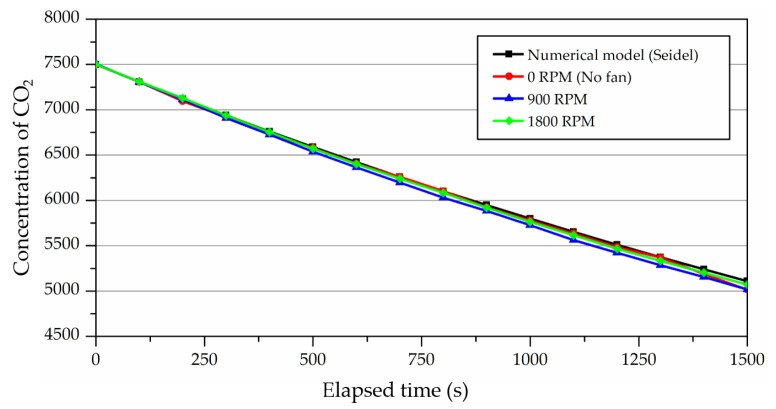
Results of ventilation experiments using tracer gas (CO_2_) according to the fan operation.

**Figure 16 ijerph-19-13556-f016:**
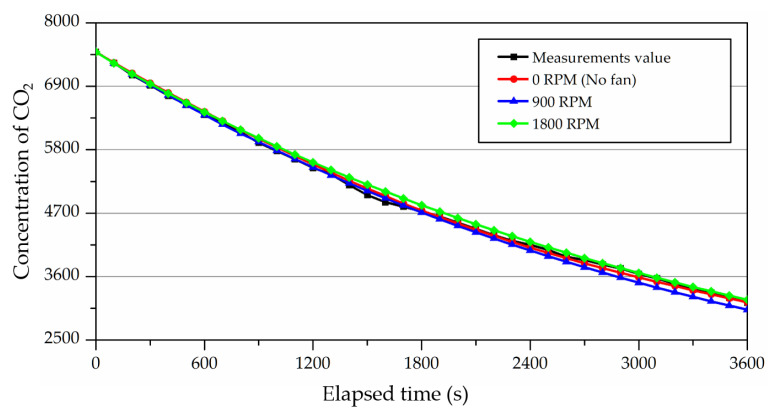
Results of CFD analysis according to the fan operation.

**Figure 17 ijerph-19-13556-f017:**
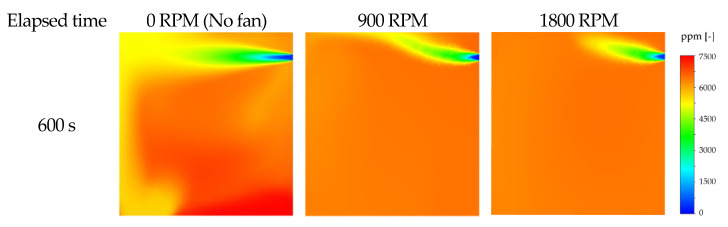
Comparison of concentration field contour according to fan operating conditions.

**Figure 18 ijerph-19-13556-f018:**
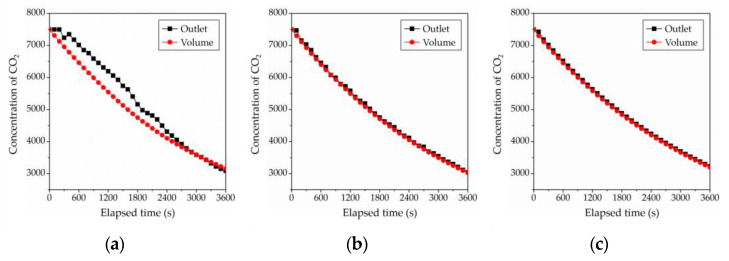
The comparison of the CO_2_ concentration decaying between outlet and volume according to fan operating conditions in CFD analysis: (**a**) 0 RPM (No fan); (**b**) 900 RPM; (**c**) 1800 RPM.

**Table 1 ijerph-19-13556-t001:** Boundary conditions for numerical analysis.

	Boundary Conditions
Cells	180,000 ± 10,000
Turbulence model	SST k-ω
Temperature	25 °C
Velocity inlet	The velocity of profile-centric: 0.6 m/s

**Table 2 ijerph-19-13556-t002:** Ventilation efficiency of five cases.

	Case 1	Case 2	Case 3	Case 4	Case 5
Age of air	3573 s	3715 s	3967 s	4299 s	3643 s
Ventilation efficiency	100.7%	96.9%	90.8%	83.7%	98.9%

**Table 3 ijerph-19-13556-t003:** Comparison of the ventilation efficiency depending on the fan operation.

	Measurements Value	CFD Analysis		
		0 RPM (No Fan)	900 RPM	1800 RPM
Age of air	3659 s	3701 s	3545 s	3779 s
Ventilation efficiency	97.7%	97.3%	101.6%	95.3%
Relative error	-	0.41%	2.47%	−3.98%

## Data Availability

Not applicable.

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
