# Peer review of "A Study of the Plan and Performance Evaluation Method of an 8-m3 Chamber Using Ventilation Experiments and Numerical Analyses"

_ijerph, 2022, doi:10.3390/ijerph192013556_

Round 1

Reviewer 1 Report

The paper is interesting and the topic is current. Thank you very much for the opportunity to study the paper and its reviews. The topic of the paper is thematically appropriate and in accordance with the focus of the journal. The paper is a clear. The paper is well structured and clear. The length of the paper is fine.  However, I consider the introduction and theoretical starting points to be insufficient. There is a lack of emphasis on the importance of the quality of the indoor environment and the effect of volatile organic substances on the quality of the indoor environment. The conclusion lacks a broader generalization in the context of the given topic. After editing, I recommend a contribution to the publication.

Many references are older than 5 or even 10 years. Here I would recommend the use of rather published knowledge not older than 5 years.  With regard to the issue, the citation apparatus could also be much more widespread. 

Comments for addition/revision: 

- Line 20: „However, no study has suggested suitable chamber design or performance evaluation methods.“ It must be rephrased and explained. There are a number of VOC test chambers (CLIMPAQ). 

- Line 42: „Indoor air pollution is often caused by formaldehyde generated by furniture or construction materials and chemical substances such as volatile organic compounds(VOCs).“ Misleading. Formaldehyde belongs to the VOCs.

- Line 44: „As a way to remove such pollutants, efforts have been made to promote the management of indoor air quality through emission experiments, and test methods and standards for regulating the emission of pollutants from furniture or construction materials have been suggested.“  Unclear. Explain IAQ management.

- Inadequate delineation against commonly available test chambers or against chambers in research facilities.

- The relationship between VOCs and CO2 is insufficiently explained. In the first paragraph there is formaldehyde, in the second it is CO2, without any explanation.

- „Figure 4. Relative error between a CFD value and measurements value of each 21 points.“ The graph in the pdf is badly cropped and contains nothing of the chart in question.

- There are some typographical errors - somewhere the subscript is missing 

- „Figure 9. The results of the ventilation experiments using tracer gas as CO2.“ It is difficult to read and it would be advisable to change the scale of the vertical axis to make the differences more visible. The same Figures 12 and 13. 

- Figure 10 is inappropriately split into two pages. 

Author Response

Dear, Reviewer 1 of IJERPH

I would like to thank you for your comments and suggestions. Careful attention has been given to incorporating the suggestions made by the reviewer.

Reviewer 2 Report

This study used both experiments and CFD simulations to evaluate ventilation efficiency in an 8-m3 chamber. CO2 was used as tracer gas. Overall, the novelty and contributions of this study needed to be further clarified. The structure of this paper was unclear. The detailed comments are as follows.

1.     The authors claimed that one of the contributions of this study used the ceiling fan to improve the ventilation efficiency due to the full mixing between indoor air and CO2. However, this practice had been recommended by the related guideline/standard, e.g., REHVA Guidebook “Ventilation effectiveness”.

2.     The structure of this paper was difficult to be understood. Especially, Sections 2 and 3 needed to be re-structured.

3.     Sections 2 and 3: Why were the standard k-e model and SST k-w used? Which model is better to predict indoor air flows?

4.     Figures 6 and 7: The numbers of the vertical axis were incorrect.

Author Response

Dear, Reviewer 2 of IJERPH

I would like to thank you for your comments and suggestions. Careful attention has been given to incorporating the suggestions made by the reviewer.

Reviewer 3 Report

In this paper, a series of experimental and numerical studies of ventilation effectiveness are conducted in an 8 m3 chamber, and the results are applied in the chamber design. Then, a method of performance evaluation is suggested by considering the imbalance in gas concentration due to the density difference of tracer gas and air. The method used in the study provides an alternative way to evaluate the indoor ventilation effectiveness. The topic is definitely worthy of investigation and is of interest to the readers of the journal. Hereafter are some comments.

1. The introduction section should be substantially strengthened. A large number of studies have been conducted with a focus on the ventilation efficiency in buildings. The key difference and gap between previous and presented study should be further clarified based on a more comprehensive review.

2. The author may consider adding a technical road-map for the reader to understand.

3. For the presented chamber, the inlet position was fixed, while the outlet position varied when studying the interior airflow of the chamber. The size and exact location of each opening should be specifically indicated, which cannot be obtained from Fig1.

4. In section 3.2, “After the injection, the fan was used to adequately mix the chamber interior air and tracer gas for 20 min to form a state of complete mixing.” How to identify the state of complete mixing can be achieved in 20 minutes?

Author Response

Dear, Reviewer 3 of IJERPH

I would like to thank you for your comments and suggestions. Careful attention has been given to incorporating the suggestions made by the reviewer.

Round 2

Reviewer 2 Report

The authors have made great efforts to improve their submission. The present manuscript is acceptable for publication. 

Author Response

I would like to thank you for your all comments and suggestions.

Reviewer 3 Report

The authors have addressed all the comments in last round.

Since the added references mainly focused on VOCs emssion test, the traditional method such as FLEC (Field and Laboratory Emission Cell) should be mentioned and the difference should be discussed as well. 

Author Response

(The authors gave the same response as above.)
